# Adaptive Sequence Submodularity

**Marko Mitrovic**
Yale University
marko.mitrovic@yale.edu

**Ehsan Kazemi**
Yale University
ehsan.kazemi@yale.edu

**Moran Feldman**
University of Haifa
moranfe@openu.ac.il

**Andreas Krause**
ETH Zürich
krausea@ethz.ch

**Amin Karbasi**
Yale University
amin.karbasi@yale.edu

## Abstract

In many machine learning applications, one needs to interactively select a sequence of items (e.g., recommending movies based on a user's feedback) or make sequential decisions in a certain order (e.g., guiding an agent through a series of states). Not only do sequences already pose a dauntingly large search space, but we must also take into account past observations, as well as the uncertainty of future outcomes. Without further structure, finding an optimal sequence is notoriously challenging, if not completely intractable. In this paper, we view the problem of adaptive and sequential decision making through the lens of submodularity and propose an adaptive greedy policy with strong theoretical guarantees. Additionally, to demonstrate the practical utility of our results, we run experiments on Amazon product recommendation and Wikipedia link prediction tasks.

## 1 Introduction

The machine learning community has long recognized the importance of both sequential and adaptive decision making. The study of sequences has led to novel neural architectures such as LSTMs [26], which have been used in a variety of applications ranging from machine translation [52] to image captioning [56]. Similarly, the study of adaptivity has led to the establishment of some of the most popular subfields of machine learning including active learning [48] and reinforcement learning [53].

In this paper, we consider the optimization of problems where both sequences and adaptivity are integral part of the process. More specifically, we focus on problems that can be modeled as selecting a sequence of items, where each of these items takes on some (initially unknown) state. The idea is that the value of any sequence depends not only on the items selected and the order of these items but also on the states of these items.

Consider recommender systems as a running example. To start, the order in which we recommend items can be just as important as the items themselves. For instance, if we believe that a user will enjoy the Lord of the Rings franchise, it is vital that we recommend the movies in the proper order. If we suggest that the user watches the final installment first, she may end up completely unsatisfied with an otherwise excellent recommendation. Furthermore, whether it is explicit feedback (such as rating a movie on Netflix) or implicit feedback (such as clicking/not clicking on an advertisement), most recommender systems are constantly interacting with and adapting to each user. It is this feedback that allows us to learn about the states of items we have already selected, as well as make inferences about the states of items we have not selected yet.

Unfortunately, the expressive modeling power of sequences and adaptivity comes at a cost. Not only does optimizing over sequences instead of sets exponentially increase the size of the search space, but adaptivity also necessitates a probabilistic approach that further complicates the problem.

Without further assumptions, even approximate optimization is infeasible. As a result, we address this challenge from the perspective of *submodularity*, an intuitive diminishing returns condition that appears in a broad scope of different areas, but still provides enough structure to make the problem tractable.

Research on submodularity, which itself has been a burgeoning field in recent years, has seen comparatively little focus on sequences and adaptivity. This is especially surprising because many problems that are commonly modeled under the framework of submodularity, such as recommender systems [21, 59] and crowd teaching [49], stand to benefit greatly from these concepts.

While the lion's share of existing research in submodularity has focused on *sets*, a few recent lines of work extend the concept of submodularity to *sequences*. Tschiatschek et al. [54] were the first to consider *sequence submodularity* in the general graph-based setting that we will follow in this paper. They presented an algorithm with theoretical guarantees for directed acyclic graphs, while Mitrovic et al. [45] developed a more comprehensive algorithm that provides theoretical guarantees for general hypergraphs.

In their experiments, both of these works showed that modeling the problem as sequence submodular (as opposed to set submodular) gave noticeable improvements. Their applications could benefit even further from the aforementioned notions of adaptivity, but the existing theory behind sequence submodularity simply cannot model the problems in this way. While adaptive *set* submodularity has been studied extensively [12, 20, 22, 24], these approaches still fail to capture order dependencies.

Alaei and Malekian [1] and Zhang et al. [60] also consider sequence submodularity (called string-submodularity in some works), but they use a different definition, which is based on subsequences instead of graphs. On the other hand, Li and Milenkovic [39] have considered the interaction of graphs and submodularity, but not in the context of sequences.

**Other Related Work** Amongst many other applications, submodularity has also been used for variable selection [36], data summarization [32, 40, 44], sensor placement [34], neural network interpretability [14], network inference [23], and influence maximization in social networks [29]. Submodularity has also been studied extensively in a wide variety of settings, including distributed and scalable optimization [5–7, 17, 18, 38, 43, 44], streaming algorithms [3, 10, 11, 19, 28, 35, 46, 47], robust optimization [9, 27, 37, 50, 55], weak submodularity [13, 15, 16, 31], and continuous submodularity [2, 4, 25, 51, 58].

**Our Contributions** The main contributions of our paper are presented in the following sections:

- In Section 2, we introduce our framework of *adaptive sequence submodularity*, which brings tractability to problems that include both sequences and adaptivity.

- In Section 3, we present our algorithm for adaptive sequence submodular maximization. We present theoretical guarantees for our approach and we elaborate on the necessity of our novel proof techniques. We also show that these techniques simultaneously improve the state-of-the-art bounds for the problem of sequence submodularity by a factor of $\frac{e}{e-1}$. Furthermore, we argue that any approximation guarantee must depend on the structure of the underlying graph unless the exponential time hypothesis is false.

- In Section 4, we use datasets from Amazon and Wikipedia to compare our algorithm against existing sequence submodular baselines, as well as state-of-the-art deep learning-based approaches.

## 2  Adaptive Sequence Submodularity

As discussed above, sequences and adaptivity are an integral part of many real-world problems. This means that many real-world problems can be modeled as selecting a sequence $\sigma$ of items from a ground set $V$, where each of these items takes on some (initially unknown) state $o \in O$. A particular mapping of items to states is known as a **realization** $\phi$, and we assume there is some unknown distribution $p(\phi)$ that governs these states.

For example in movie recommendation, the set of all movies is our ground set $V$ and our goal is to select a sequence of movies that a particular user will enjoy. If we recommend a movie $v_i \in V$ and the user likes it, we place $v_i$ in state 1 (i.e. $o_i = 1$). If not, we put it into state 0. Naturally, the value of a movie should be higher if the user liked it, and lower if she did not.

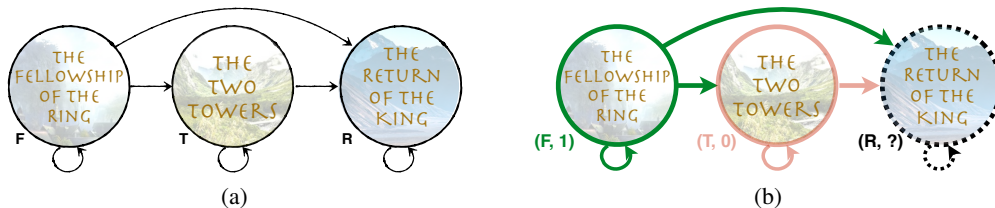

(a)                                             (b)

Figure 1: (a) shows an underlying graph for a movie recommendation problem. The vertices are movies and edges denote the additional value of watching certain movies in certain orders. (b) extends this to the adaptive case, where both the vertices and the edges take on a state. The user has reported that she liked the Fellowship of the Ring (so it is placed in state 1), but she did not like The Two Towers (so it is placed in state 0). The state of the last movie is still unknown. In this example, the state of an edge is equal to the state of its starting vertex.

Formally, we want to select a sequence $\sigma$ that maximizes $f(\sigma, \phi)$, where $f(\sigma, \phi)$ is the value of sequence $\sigma$ under realization $\phi$. However, $\phi$ is initially unknown to us and the state of each item in the sequence is revealed to us only after we select it. In fact, even if we knew $\phi$ perfectly, the set of all sequences poses an intractably large search space. From an optimization perspective, this problem is hopeless without further structural assumptions.

Our first step towards taming this problem is to follow the work of Tschiatschek et al. [54] and assume that the value of a sequence can be defined using a graph. Concretely, we have a directed graph $G = (V, E)$, where each item in our ground set is represented as a vertex $v \in V$, and the edges encode the additional value intrinsic to picking certain items in certain orders. Mathematically, selecting a sequence of items $\sigma$ will induce a set of edges $E(\sigma)$:

$$E(\sigma) = \big\{ (\sigma_i, \sigma_j) \mid (\sigma_i, \sigma_j) \in E, i \leq j \big\}.$$

For example, consider the graph in Figure 1a and consider the sequence $\sigma_A = [F, T]$ where the user watched The Fellowship of the Ring, and then The Two Towers, as well as the sequence $\sigma_B = [T, F]$ where the user watched the same two movies but in the opposite order.

$E(\sigma_A) = E\big([F, T]\big) = \big\{ (F, F), (T, T), (F, T) \big\}$
$E(\sigma_B) = E\big([T, F]\big) = \big\{ (T, T), (F, F) \big\}$

Using the self-loops, this graph encodes the fact that there is certainly some intrinsic value to watching these movies regardless of the order. On the other hand, the edge $(F, T)$ encodes the fact that watching The Fellowship of the Ring before The Two Towers will bring additional value to the viewer, and this edge is only induced if the movies appear in the correct order in the sequence.

With this graph based set-up, however, we run into issues when it comes to adaptivity. In particular, the states of items naturally translate to states for the vertices, but it is not clear how to extend adaptivity to the *edges*. We tackle this challenge by assigning a state $q \in Q$ to each edge strictly as a function of the states of its endpoints. That is, similarly to how a sequence $\sigma$ induces a set of edges $E(\sigma)$, a realization $\phi$ for the states of the vertices induces a realization $\phi^E$ for the states of the edges. We want to emphasize that our framework works for *any* deterministic mapping from vertex states to edge states. One simple option that we will use throughout this paper as a running example is to define the state of an edge to always be equal to the state of its start vertex.

As we will discuss later, the analysis for this approach will necessitate some novel proof techniques, but the resulting framework is very flexible and it allows us to fully redefine the adaptive sequence problem in terms of the underlying graph:

$$f(\sigma, \phi) = h\big(E(\sigma), \phi^E\big) \text{ where } \sigma \text{ induces } E(\sigma) \text{ and } \phi \text{ induces } \phi^E.$$

The last necessary ingredient to bring tractability to this problem is submodularity. In particular, we will assume that $h\big(E(\sigma), \phi^E\big)$ is *weakly adaptive set submodular*. This is a relaxed version of standard adaptive set submodularity that can model an even larger variety of problems, and it is a natural fit for the applications we consider in this paper.

In order to formally define weakly-adaptive submodularity, we need a bit more terminology. To start, we define a **partial realization** $\psi$ to be a mapping for only some subset of items (i.e., the states of

the remaining items are unknown). For notational convenience, we define the domain of $\psi$, denoted $dom(\psi)$, to be the list of items $v$ for which the state of $v$ is known. We say that $\psi$ is a **subrealization** of $\psi'$, denoted $\psi \subseteq \psi'$, if $dom(\psi) \subseteq dom(\psi')$ and they are equal everywhere in the domain of $\psi$. Intuitively, if $\psi \subseteq \psi'$, then $\psi'$ has all the same information as $\psi$, and potentially more.

Given a partial realization $\psi$, we define the marginal gain of a set $A$ as

$$\Delta(A \mid \psi) = \mathbb{E}\Big[h\big(dom(\psi) \cup A, \phi\big) - h\big(dom(\psi), \phi\big) \mid \psi\Big],$$

where the expectation is taken over all full realizations $\phi$ such that $\psi \subseteq \phi$. In other words, we condition on the states given by the partial realization $\psi$, and then we take the expectation across all possibilities for the remaining states.

**Definition 1.** *A function $h : 2^E \times Q^E \to \mathbb{R}_{\geq 0}$ is **weakly adaptive set submodular** with parameter $\gamma$ if for all sets $A \subseteq E$ and for all $\psi \subseteq \psi'$ we have:*

$$\Delta(A \mid \psi') \leq \frac{1}{\gamma} \cdot \sum_{e \in A} \Delta(e \mid \psi).$$

This notion is a natural generalization of weak submodular functions [13] to adaptivity. The primary difference is that we condition on subrealizations instead of just sets because we need to account for the states of items. Note that in the context of this paper $h$ is a function on the edges, so we will condition on subrealizations of the edges $\psi^E$. However, these concepts apply more generally to functions on any set and state spaces, so we use $\psi$ in the formal definitions.

**Definition 2.** *A function $h : 2^E \times Q^E \to \mathbb{R}_{\geq 0}$ is **adaptive monotone** if $\Delta(e \mid \psi) \geq 0$ for all partial realizations $\psi$. That is, the conditional expected marginal benefit of any element is non-negative.*

Figure 1b is designed to help clarify these concepts. It includes the same graph as Figure 1a, but now we can receive feedback from the user. If we recommend a movie and the user likes it, we put the corresponding vertex in state 1 (green in the image). Otherwise, we put the vertex in state 0 (red in the image). Vertices whose states are still unknown are denoted by a dotted black line.

Next, in our example, we need to define a state for each edge in terms of the states of its endpoints. In this case, we will define the state of each edge to be equal to the state of its start point. In Figure 1b, the user liked The Fellowship of the Ring, which puts edges $(F, F)$, $(F, T)$, and $(F, R)$ in state 1 (green). She did not like The Two Towers, so edges $(T, T)$ and $(T, R)$ are in state 0 (red), and we do not know the state for The Return of the King, so the state of $(R, R)$ is also unknown. We call this partial realization $\psi_1$ for the vertices, and the induced partial realization for the edges $\psi_1^E$.

Suppose our function $h$ counts all induced edges that are in state 1. Furthermore, let us simply assume that any unknown vertex is equally likely to be in state 0 or state 1. This means that the self-loop $(R, R)$ is also equally likely to be in either state 0 or state 1. Therefore, $\Delta\big((R, R) \mid \psi_1^E\big) = \frac{1}{2} \times 0 + \frac{1}{2} \times 1 = \frac{1}{2}$.

On the other hand, consider the edge $(F, R)$. Under $\psi_1$, we know $F$ is in state 1, which means $(F, R)$ is also in state 1, and thus, $\Delta\big((F, R) \mid \psi_1^E\big) = 1$. However, if we consider a subrealization $\psi_2 \subseteq \psi_1$ where we do not know the state of $F$, then it is equally likely to be in either state and $\Delta\big((F, R) \mid \psi_2^E\big) = \frac{1}{2} \times 0 + \frac{1}{2} \times 1 = \frac{1}{2}$. Therefore, for this simple function we know that $\gamma \leq 0.5$.

## 3   Adaptive Sequence-Greedy Policy and Theoretical Results

In this section, we introduce our Adaptive Sequence-Greedy policy and present its theoretical guarantees. We first formally define **weakly adaptive sequence submodularity**.

**Definition 3.** *A function $f(\sigma, \phi)$ defined over a graph $G(V, E)$ is **weakly adaptive sequence submodular** if $f(\sigma, \phi) = h\big(E(\sigma), \phi^E\big)$ where a sequence $\sigma$ of vertices in $V$ induces a set of edges $E(\sigma)$, realization $\phi$ induces $\phi^E$, and the function $h$ is weakly adaptive set submodular. Note that if $h$ is adaptive monotone, then $f$ is also adaptive monotone.*

Formally, a policy $\pi$ is an algorithm that builds a sequence of $k$ vertices by seeing which states have been observed at each step, then deciding which vertex should be chosen and observed next. If $\sigma_{\pi, \phi}$ is the sequence returned by policy $\pi$ under realization $\phi$, then we write the expected value of $\pi$ as:

$$f_{\text{avg}}(\pi) = \mathbb{E}\big[f(\sigma_{\pi, \phi}, \phi)\big] = \mathbb{E}\Big[h\big(E(\sigma_{\pi, \phi}), \phi^E\big)\Big]$$

where again the expectation is taken over all possible realizations $\phi$. Our goal is to find a policy $\pi$ that maximizes $f_{\mathrm{avg}}(\pi)$, as defined above.

Our Adaptive Sequence Greedy policy $\pi$ (Algorithm 1) starts with an empty sequence $\sigma$. Throughout the policy, we define $\psi_\sigma$ to be the partial realization for the vertices in $\sigma$. In turn this gives us the partial realization $\psi_\sigma^E$ for the induced edges.

At each step, we define the valid set of edges $\mathcal{E}$ to be the edges whose endpoint is not already in $\sigma$. The main idea of our policy is that, at each step, we select the valid edge $e \in \mathcal{E}$ with the highest expected value $\Delta(e \mid \psi_\sigma^E)$. For each such edge, the endpoints that are not already in the sequence $\sigma$ are concatenated ($\oplus$ means concatenate) to the end of $\sigma$, and their states are observed (updating $\psi_\sigma$).

---

**Algorithm 1** Adaptive Sequence Greedy Policy $\pi$

1: **Input:** Directed graph $G = (V, E)$, weakly adaptive sequence submodular $f(\sigma, \phi) = h\big(E(\sigma), \phi^E\big)$, and cardinality constraint $k$
2: Let $\sigma \leftarrow ()$
3: **while** $|\sigma| \leq k - 2$ **do**
4: $\quad \mathcal{E} = \{e_{ij} \in E \mid v_j \notin \sigma\}$
5: $\quad$ **if** $\mathcal{E} \neq \varnothing$ **then**
6: $\quad\quad e_{ij} = \arg\max_{e \in \mathcal{E}} \Delta(e \mid \psi_\sigma^E)$
7: $\quad\quad$ **if** $v_i = v_j$ **or** $v_i \in \sigma$ **then**
8: $\quad\quad\quad \sigma = \sigma \oplus v_j$ and observe state of $v_j$
9: $\quad\quad$ **else**
10: $\quad\quad\quad \sigma = \sigma \oplus v_i \oplus v_j$ and observe states of $v_i, v_j$
11: $\quad\quad$ **end if**
12: $\quad$ **else**
13: $\quad\quad$ **break**
14: $\quad$ **end if**
15: **end while**
16: **Return** $\sigma$

---

**Theorem 1.** *For adaptive monotone and weakly adaptive sequence submodular function $f$, the Adaptive Sequence Greedy policy $\pi$ represented by Algorithm 1 achieves*

$$f_{avg}(\pi) \geq \frac{\gamma}{2d_{\mathrm{in}} + \gamma} \cdot f_{avg}(\pi^*),$$

*where $\gamma$ is the weakly adaptive submodularity parameter, $\pi^*$ is the policy with the highest expected value and $d_{\mathrm{in}}$ is the largest in-degree of the input graph $G$.*

As discussed by Mitrovic et al. [45], using a hypergraph $H$ instead of a normal graph $G$ allows us to encode more intricate relationships between the items. For example, in Figure 1a, the edges only encode pairwise relationships. However, there may be relationships between larger groups of items that we want to encode explicitly. For instance, if included, the value of a hyperedge $(F, T, R)$ in Figure 1a would explicitly encode the value of watching The Fellowship of the Ring, followed by watching The Two Towers, and then concluding with The Return of the King.

We can also extend our policy to general hypergraphs (see Algorithm 2 in Appendix B.3). Theorem 2 guarantees the performance of our proposed policy for hypergraphs.

**Theorem 2.** *For adaptive monotone and weakly adaptive sequence submodular function $f$, the policy $\pi'$ represented by Algorithm 2 achieves*

$$f_{avg}(\pi') \geq \frac{\gamma}{rd_{\mathrm{in}} + \gamma} \cdot f_{avg}(\pi^*),$$

*where $\gamma$ is the weakly adaptive submodularity parameter, $\pi^*$ is the policy with the highest expected value and $r$ is the size of the largest hyperedge in the input hypergraph.*

In our proofs, we have to handle the sequential nature of picking items and the revelation of states in a combined setting. Unfortunately, the existing proof methods for sequence submodular maximization are not linear enough to allow for the use of the linearity of expectation that captures the stochasticity of the states. For this reason, we develop a novel analysis technique to guarantee the performance

of our algorithms. Our proof replaces several lemmas from Mitrovic et al. [45] with tighter, more linear analyses. Surprisingly, these new techniques also improve the theoretical guarantees of the non-adaptive Sequence-Greedy and Hyper Sequence-Greedy [45] by a factor of $\frac{e}{e-1}$.

**Proofs for both theorems are given in Appendix B.**

**General Unifying Framework** One more theoretical point we want to highlight is that weakly adaptive sequence submodularity provides a general unifying framework for a variety of common submodular settings including, adaptive submodularity, weak submodularity, sequence submodularity, and classical set submodularity. If we have $\gamma = 1$ and the state of all vertices is deterministic, then we have sequence submodularity. Conversely, if the vertex states are unknown, but our graph only has self-loops, then we have weakly adaptive set submodularity (and correspondingly adaptive set submodularity if $\gamma = 1$). Lastly, if we have a graph with only self-loops, full knowledge of all states, and $\gamma = 1$, then we recover the original setting of classical set submodularity.

**Tightness of Theoretical Results** We acknowledge that the constant factor approximation we present depends on the maximum in-degree. While ideally the theoretical bound would be completely independent of the structure of the graph, we argue here that such a dependence is likely necessary.

Indeed, getting a dependence better than $O(n^{1/4})$ in the approximation factor (where $n$ is the total number of items) would improve the state-of-the-art algorithm for the very well-studied densest $k$ subgraph problem (DkS) [8, 33]. Moreover, if we could get an approximation that is completely independent of the structure of the graph, then the exponential time hypothesis would be proven false[1]. In fact, even an almost polynomial approximation would break the exponential time hypothesis [41]. Next, we formally state this hardness relationship. **The proof is given in Appendix C**.

**Theorem 3.** *Assuming the exponential time hypothesis is correct, there is no algorithm that approximates the optimal solution for the (adaptive) sequence submodular maximization problem within a $n^{1/(\log \log n)^c}$ factor, where $n$ is the total number of items and $c > 0$ is a universal constant independent of $n$.*

## 4 Experimental Results

### 4.1 Amazon Product Recommendation

Using the Amazon Video Games review dataset [42], we consider the task of recommending products to users. In particular, given the first $g$ products that the user has purchased, we want to predict the next $k$ products that she will buy. Full experimental details are given in Appendix D.1. **Dataset and code are attached in the supplementary material.**

We start by using the training data to build a graph $G = (V, E)$, where $V$ is the set of all products and $E$ is the set of edges between these products. The weight of each edge, $w_{ij}$, is defined to be the conditional probability of purchasing product $j$ given that the user has previously purchased product $i$. There are also self-loops with weight $w_{ii}$ that represent the fraction of users that purchased product $i$.

We define the state of each edge $(i, j)$ to be equal to the state of product $i$. The intuitive idea is that edge $(i, j)$ encodes the value of purchasing product $j$ after already having purchased product $i$. Therefore, if the user has definitely purchased $i$ (i.e., product $i$ is in state 1), then they should receive the full value of $w_{ij}$. On the other hand, if she has definitely not purchased $i$ (i.e., product $i$ is in state 0), then edge $(i, j)$ provides no value. Lastly, if the state of $i$ is unknown, then the expected gain of edge $(i, j)$ is discounted by $w_{ii}$, the value of the self-loop on $i$, which can be viewed as a simple estimate for the probability of the user purchasing product $i$. See Figure 2a for a small example.

We use a probabilistic coverage utility function as our monotone weakly-adaptive set submodular function $h$. Mathematically,

$$h(E_1) = \sum_{j \in V} \left[ 1 - \prod_{(i,j) \in E_1} (1 - w_{ij}) \right],$$

where $E_1 \subseteq E$ is the subset of edges that are in state 1. Note that with this set-up, the value of $\gamma$ can be difficult to calculate exactly. However, roughly speaking, it is inversely proportional to the value of the smallest weight self-loop $w_{ii}$.

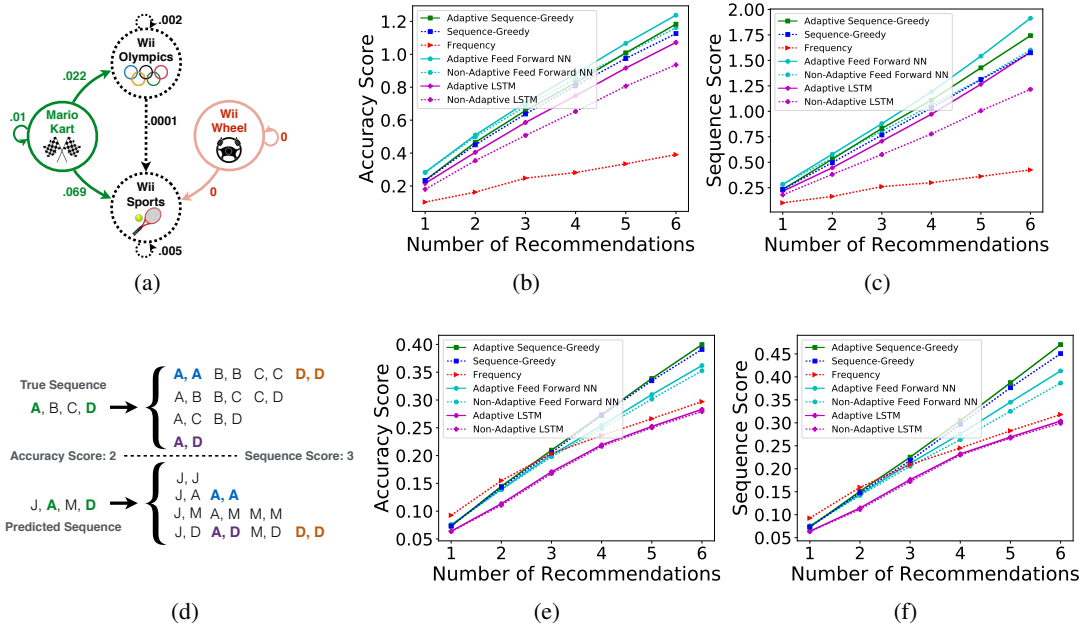

Figure 2: (a) shows a small subset of the underlying graph with states for a particular user. (b) and (c) show our results on the Amazon product recommendation task. In all these graphs, the number of given products $g$ is 4. (d) gives an example illustrating the difference between the two performance measures. (e) and (f) show our results on the same task, but using only 1% of the available training to show that our algorithm outperforms deep learning-based approaches in data scarce environments.

We compare the performance of our Adaptive Sequence-Greedy policy against Sequence-Greedy from Mitrovic et al. [45], the existing sequence submodularity baseline that does not consider states. To give further context for our results, we compare against Frequency, a naive baseline that ignores sequences and adaptivity and simply outputs the $k$ most popular products.

We also compare against a set of deep learning-based approaches (see Appendix D.3 for full details). In particular, we implement adaptive and non-adaptive versions of both a regular Feed Forward Neural Network and an LSTM. The adaptive version will update its inputs after every prediction to reflect whether or not the user liked the recommendation. Conversely, the non-adaptive version will simply make $k$ predictions using just the original input.

We use two different measures to compare the various algorithms. The first is the **Accuracy Score**, which simply counts the number of recommended products that the user indeed ended up purchasing. While this is a sensible measure, it does not explicitly consider the order of the sequence. Therefore, we also consider the **Sequence Score**, which is a measure based on the Kendall-Tau distance [30]. In short, this measure counts the number of ordered pairs that appear in both the predicted sequence and the true sequence. Figure 2d gives an example comparing the two measures.

Figures 2b and 2c show the performance of the various algorithms using the accuracy score and sequence score, respectively. These results highlight the importance of adaptivity as the adaptive algorithms consistently outperform their non-adaptive counterparts under both scoring regimes. Notice that in both cases, as the number of recommendations increases, our proposed Adaptive Sequence-Greedy policy is outperformed only by the Adaptive Feed Forward Neural Network. Although LSTMs are generally considered better for sequence data than vanilla feed-forward networks, we think it is a lack of data that causes them to perform poorly in our experiments.

Another observation, which fits the conventional wisdom, is that deep learning-based approaches can perform well when there is a lot of data. However, when the data is scarce, we see that the Sequence-Greedy based approaches outperform the deep learning-based approaches. Figures 2e and 2f simulate a data-scarce environment by using only 1% of the available data as training data. Note that the difference between the adaptive algorithms and their non-adaptive counterparts is

less obvious in this setting because the adaptive algorithms use correct guesses to improve future recommendations, but the data scarcity makes it difficult to make a correct guess in the first place.

Aside from competitive accuracy and sequence scores, the Adaptive Sequence-Greedy algorithm provides several advantages over the neural network-based approaches. From a theoretical perspective, the Adaptive Sequence-Greedy algorithm has provable guarantees on its performance, while little is known about the theoretical performance of neural networks. Furthermore, the decisions made by the Adaptive Sequence-Greedy algorithm are easily interpretable and understandable (it is just picking the edge with the highest expected value), while neural networks are generally a black-box. On a similar note, Adaptive Sequence-Greedy may be preferable from an implementation perspective because it does not require any hyperparameter tuning. It is also more robust to changing inputs in the sense that we can easily add another product and its associated edges to our graph, but adding another product to the neural network requires changing the entire input and output structure, and thus, generally necessitates retraining the entire network.

### 4.2 Wikipedia Link Prediction

Using the Wikispeedia dataset [57], we consider users who are surfing through Wikipedia towards some target article. Given a sequence of articles the user has previously visited, we want to guide her to the page she is trying to reach. Since different pages have different valid links, the order of pages we visit is critical to this task. Formally, given the first $g = 3$ pages each user visited, we want to predict which page she is trying to reach by making a series of suggestions for which link to follow.

In this case, we have $G = (V, E)$, where $V$ is the set of all pages and $E$ is the set of existing links between pages. Similarly to before, the weight $w_{ij}$ of an edge $(i, j) \in E$ is the probability of moving to page $j$ given that the user is currently at page $i$. In this case, there are no self-loops as we assume we can only move using links, and thus we cannot jump to random pages. We again define two states for the nodes: 1 if the user definitely visits this page and 0 if the user does *not* want to visit this page.

This application highlights the importance of adaptivity because the non-adaptive sequence submodularity framework cannot model this problem properly. This is because the Sequence-Greedy algorithm is free to choose any edge in the underlying graph, so there is no way to force the algorithm to pick a link that is connected to the user's current page. On the other hand, with Adaptive Sequence-Greedy, we can use the states to penalize invalid edges, and thus force the algorithm to select only links connected to the user's current page. Similarly, we only have the adaptive versions of the deep learning baselines because we need information about our current page in order to construct a valid path (Appendix D.3 gives a more detailed explanation).

Figure 3a shows an example of predicted paths, while Figure 3b shows our quantitative results. More detail about the relevance distance metric is given in Appendix D.2, but the idea is that the it measures the relevance of the final output page to the true target page (a lower score indicates a higher relevance). The main observation here is that the Adaptive Sequence Greedy algorithm actually outperforms the deep-learning based approaches. The main reason for this discrepancy is likely a lack of data as we have 619 pages to choose from and only 7,399 completed search paths.

## 5 Conclusion

In this paper we introduced adaptive sequence submodularity, a general framework for bringing tractability to the broad class of optimization problems that consider both sequences and adaptivity. We presented Adaptive Sequence-Greedy—a general policy for optimizing weakly adaptive sequence submodular functions. We provide a provable theoretical guarantee for our algorithm, as well as a discussion about the tightness of our result. Our novel analysis also improves the theoretical guarantees of Sequence-Greedy and Hyper Sequence-Greedy [45] by a factor of $\frac{e}{e-1}$. Finally, we evaluated the performance of Adaptive Sequence-Greedy on an Amazon product recommendation task and a Wikipedia link prediction task. Not only does our Adaptive Sequence-Greedy policy exhibit competitive performance with the state-of-the-art, but it also provides several notable advantages, including interpretability, ease of implementation, and robustness against both data scarcity and input adjustments.

**Acknowledgements.** This work was partially supported by NSF (IIS-1845032), ONR (N00014-19-1-2406), AFOSR (FA9550-18-1-0160), ISF (1357/16), and ERC StG SCADAPT.

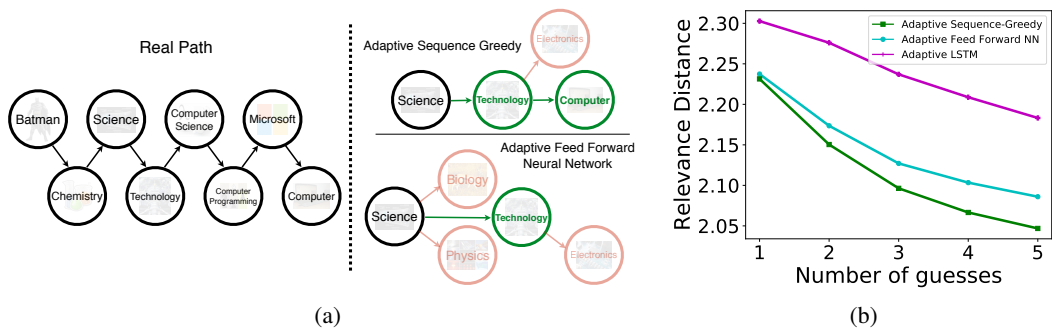

<div align="center">(a)          (b)</div>

Figure 3: (a) The left side shows the real path a user followed from *Batman* to *Computer*. Given the first three pages, the right side shows the path predicted by Adaptive Sequence Greedy versus a deep learning-based approach. Green shows correct guesses that were followed, while red shows incorrect guesses that were not pursued further. (b) shows the overall performance of the various approaches.

## Footnotes

[1]If the exponential time hypothesis is true it would imply that P $\neq$ NP, but it is a stronger statement.

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
