[Supplementary Material]

# Adaptive Sequence Submodularity

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

[2]Note that there is a one to one correspondence between a realization $\phi$ over the vertices and a realization $\phi^E$ over the edges.

[3]Note that in this section we define the approximation factor as the ratio of the the optimal solution to the solution provided by the algorithm.

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

## A  Table of Notations

Table 1

| $E$ | Ground set of elements. |
|---|---|
| $e \in E$ | An individual element from $E$. |
| $\phi$ | A realization, i.e., a function from elements to states. |
| $\psi$ | A partial realization to encoding the current set of observations. |
| $\mathrm{dom}(\psi)$ | Domain of a partial realization $\psi$ is defined as $\mathrm{dom}(\psi) = \{e : \exists o \text{ .s.t. } (o, e) \in \psi\}$. |
| $\Phi, \Psi$ | A random realization and a random partial realization, respectively. |
| $\sim$ | For a realization $\phi$ and a partial realization $\psi$: $\phi \sim \psi$ means $\psi(e) = \phi(e)$ for all $e \in \mathrm{dom}(\psi)$. |
| $p(\phi)$ | The probability distribution on realizations. |
| $p(\phi \mid \psi)$ | The conditional distribution on realizations: $p(\phi \mid \psi) \triangleq \Pr[\Phi = \phi \mid \Phi \sim \psi]$. |
| $\pi$ | A policy, which maps partial realizations to items. |
| $E(\pi, \phi)$ | The set of all edges induced by $\pi$ when run under realization $\phi$. |
| $h$ | An objective function of type $h : 2^E \times O^E \to \mathbb{R}_{\geq 0}$. |
| $\Delta(e \mid \psi)$ | The conditional expected marginal benefit of $e$ conditioned on $\psi$. |
| $k$ | Budget on the number of selected items. |

## B  Proofs

In this section, we prove Theorems 1 and 2. Towards this goal, we first state some necessary definitions and notations, and present a few results regarding weakly adaptive submodular functions.

### B.1  Weakly Adaptive Sequence Submodular

**Notation**  The random variable $\Phi$ denotes a random realization with respect to the distribution $p(\Phi = \phi)$ over the items (or equivalently vertices of the graph).[2] For a set $A$, its partial realization (i.e., items in $A$ and their corresponding states) is shown by $\psi_A = \{(e, O(e)) \mid e \in A\}$, where $O(e)$ gives the state of $e$. For a partial realization $\psi$, we define $\mathrm{dom}(\psi) = \{e : \exists o \text{ s.t. } (o, e) \in \psi\}$. We use $\Psi_A$ to denote a random partial realization over a set $A$. Note that the distribution of random variable $\Psi_A$ is uniquely defined by the distribution of random variable $\Phi$. A partial realization $\psi$ is consistent with a realization $\phi$ (we write $\phi \sim \psi$) if they are equal, i.e., they are in the same state, everywhere in the domain of $\psi$. For the ease of notation, we define $h(\psi) \triangleq h(\mathrm{dom}(\psi), O(\psi))$, where $O(\psi)$ is the state of items in the realization $\psi$. We also define $h_{avg}(A) \triangleq \mathbb{E}_\Phi(h(A)) \triangleq \mathbb{E}_\Phi[h(\Phi_A)]$ which is the expected utility of set $A$ (and states of its elements) over all possible realizations of $A$ under the probability distribution $p(\Phi = \phi)$. We define $\Delta(e \mid \psi) = \mathbb{E}_{\Phi \sim \psi}[h(\Psi_{\{e\}} + \psi) - h(\psi)]$ which is the conditional expected marginal benefit of item $e$ conditioned on having observed the subrealization $\psi$. Note that the random variable $\Psi_{\{e\}}$ is the state of item $e$ with respect to the probability distribution $p(\Phi = \phi \mid \Phi \sim \psi)$. Similarly, we define $\Delta(A \mid \psi) = \mathbb{E}_{\Phi \sim \psi}[h(\Psi_A + \psi) - h(\psi)]$ which is the expected marginal gain of set $A$ to the partial realization $\psi$. Assume $E(\pi_\phi)$ is the set of edges induced by the set of items policy $\pi$ selects under the realization $\phi$. The expected utility of policy $\pi$ is defined as $f_{avg}(\pi) \triangleq h_{avg}(E(\pi)) = \mathbb{E}_\Phi[h(E(\pi_\Phi)]$, where the expectation is taken with respect to $p(\Phi = \phi)$. For a list of all the notations used in the paper refer to Table 1 in Appendix A.

Next, we restate the definitions for weakly adaptive set submodular and adaptive monotone functions.

**Definition 1.** *A function $h : 2^E \times Q^E \to \mathbb{R}_{\geq 0}$ is **weakly adaptive set submodular** with parameter $\gamma$ if for all sets $A \subseteq E$ and for all $\psi \subseteq \psi'$ we have:*

$$\Delta(A \mid \psi') \leq \frac{1}{\gamma} \cdot \sum_{e \in A} \Delta(e \mid \psi).$$

**Definition 2.** *A function $h : 2^E \times Q^E \to \mathbb{R}_{\geq 0}$ is **adaptive monotone** if $\Delta(e \mid \psi) \geq 0$ for all partial realizations $\psi$. That is, the conditional expected marginal benefit of any element is non-negative.*

485 Definition 1 is the generalization of both weak submodularity [13] and adaptive submodularity [22]
486 concepts.

487 Next, we state a few useful claims regarding weakly adaptive submodular functions.

488 First note for all $\psi$ and for every set $A \subseteq E \setminus \text{dom}(\psi)$, from Definition 1 and the fact that $\psi \subseteq \psi$, we
489 have

$$\Delta(A \mid \psi) \leq \frac{1}{\gamma} \cdot \sum_{e \in A} \Delta(e \mid \psi). \tag{1}$$

490 **Lemma 1.** *For all $\psi$ and $A \subseteq B \subseteq E \setminus dom(\psi)$, we have*

$$\Delta(B \mid \psi) - \Delta(A \mid \psi) \leq \frac{1}{\gamma} \cdot \sum_{e \in B \setminus A} \Delta(e \mid \psi).$$

491 *Proof.* We have

$$\Delta(B \mid \psi) - \Delta(A \mid \psi) = \sum \Pr[\Psi_A = \psi' \mid \Phi \sim \psi] \cdot \sum \Pr[\Psi_{B \setminus A} = \psi'' \mid \Phi \sim \psi + \psi']$$
$$\cdot (h_{avg}(\psi + \psi' + \psi'') - h_{avg}(\psi + \psi'))$$
$$= \sum \Pr[\Psi_A = \psi' \mid \Phi \sim \psi] \cdot \Delta(B \setminus A \mid \psi + \psi') \leq \frac{1}{\gamma} \cdot \sum_{e \in B \setminus A} \Delta(e \mid \psi),$$

492 where the inequality is derived from the definition of weakly adaptive set submodular functions (see
493 Definition 1) and the fact that $\sum \Pr[\Psi_A = \psi' \mid \Phi \sim \psi] = 1$. □

494 **Corollary 1.** *For all $\psi$, $e^* = \arg\max_{e \in E} \Delta(e \mid \psi)$ and two random subsets $A \subseteq B \subseteq E \setminus dom(\psi)$*
495 *whose randomness might depend on the realization, we have*

$$\mathbb{E}[\Delta(B \mid \psi) - \Delta(A \mid \psi) \mid \Phi \sim \psi] \leq \frac{\mathbb{E}[|B \setminus A| \mid \Phi \sim \psi]}{\gamma} \cdot \Delta(e^* \mid \psi).$$

496 *Proof.* By taking expectation over the guarantee of Lemma 1, we get

$$\mathbb{E}[\Delta(B \mid \psi) - \Delta(A \mid \psi) \mid \Phi \sim \psi] \leq \frac{1}{\gamma} \cdot \mathbb{E}\left[ \sum_{e \in B \setminus A} \Delta(e \mid \psi) \mid \Phi \sim \psi \right]$$
$$\leq \frac{1}{\gamma} \cdot \mathbb{E}\left[ \sum_{e \in B \setminus A} \Delta(e^* \mid \psi) \mid \Phi \sim \psi \right]$$
$$= \frac{\mathbb{E}[|B \setminus A| \mid \Phi \sim \psi]}{\gamma} \cdot \Delta(e^* \mid \psi),$$

497 where the second inequality follows from the fact that $e^*$ is the element with the largest expected
498 gain. □

499 The following observation is an immediate consequence of Definition 2.

500 **Observation 1.** *For any two (possibly random) subsets $A \subseteq B \subseteq E$, we have*

$$\mathbb{E}_\Phi(h(A)) \leq \mathbb{E}_\Phi(h(B)).$$

501 **Lemma 2.** *Assume $h$ is adaptive monotone and weakly adaptive set submodular with a parameter $\gamma$*
502 *with respect to the distribution $p(\phi)$, and $\pi$ is a greedy policy which picks the item with the largest*
503 *expected marginal gain at each step, then for all policies $\pi^*$ we have*

$$h_{avg}(\pi) \geq \left(1 - e^{-1/\gamma}\right) \cdot h_{avg}(\pi^*).$$

504 *Proof.* The proof of this lemma follows the same line of argument as the proof of [22, Theroem 5].
505 □

## B.2   Proof of Theorem 1

In this section, we first restate Theorem 1 and then prove it.

**Theorem 1.** *For adaptive monotone and weakly adaptive sequence submodular function $f$, the Adaptive Sequence Greedy policy $\pi$ represented by Algorithm 1 achieves*

$$f_{avg}(\pi) \geq \frac{\gamma}{2d_{\text{in}} + \gamma} \cdot f_{avg}(\pi^*),$$

*where $\gamma$ is the weakly adaptive submodularity parameter, $\pi^*$ is the policy with the highest expected value and $d_{\text{in}}$ is the largest in-degree of the input graph $G$.*

We assume the function $h$ is weakly adaptive set submodular (with a parameter $\gamma$) and monotone adaptive submodular. Furthermore, we assume $\pi^*$ is the optimal policy. It means $\pi^*$ maximizes the expected gain over the distribution $\Phi$.

Let $\ell = \lceil k/2 \rceil$. For every $0 \leq s \leq \ell$, let $\pi_s$ be the set of items picked by the greedy policy $\pi$ after $s$ iterations (if the algorithm does not make that many iterations because the set $\mathcal{E}$ became empty at some earlier point, then we assume for the sake of the proof that the algorithm continues to make dummy iterations after the point in which $\mathcal{E}$ becomes empty, and in the dummy iterations it picks no items). The observed partial realization of edges after $s$ iterations of the algorithm is represented by $\psi_s$. The random variable representing $\psi_s$ is $\Psi_s$. We define $f_{avg}(\pi_s) \triangleq h_{avg}(E(\pi_s))$, i.e., it is the expected value of items picked by the greedy policy $\pi$ after $s$ iterations. For every $1 \leq s \leq \ell$, we also denote by $e_s$ and $\mathcal{E}_s$ the values assigned to the variables $e_{ij}$ and $\mathcal{E}$, respectively, at iteration number $s$. Finally, we assume $e_s$ is a dummy arc with zero marginal contribution to $h$ if iteration number $s$ is a dummy iteration (i.e., the algorithm makes in reality less than $s$ iterations).

**Observation 2.** *For every $0 \leq s_1 \leq s_2 \leq \ell$, conditioned on the partial realization $\psi_{s_1}$, i.e., the policy has already made its first $s_1$ iterations, we have $\mathcal{E}_{s_1} \supseteq \mathcal{E}_{s_2}$ and $E(\pi_{s_1}) \subseteq E(\pi_{s_2})$.*

*Proof.* Both properties guaranteed by the observation follow from the fact that: for all possible realization $\phi \sim \psi_{s_1}$, we have that $\pi_{s_1}$ is a (possibly trivial) prefix of $\pi_{s_2}$. $\qquad\square$

**Lemma 3.** *For every $1 \leq s \leq \ell$, $f_{avg}(\pi_s) - f_{avg}(\pi_{s-1}) \geq \mathbb{E}_{\Psi_{s-1}}[\Delta(e_s \mid \Psi_{s-1})]$.*

*Proof.* Consider a fixed sub-realization $\psi_{s-1}$. If $e_s$ is a dummy arc, then $\pi_s = \pi_{s-1}$, and the observation is trivial. Otherwise, notice that the membership of $e_s$ in $\mathcal{E}_{s-1}$ guarantees that it does not belong to $E(\pi_{s-1}) = \text{dom}(\psi_{s-1})$, but does belong to $E(\sigma_s)$. Together with the fact that $E(\pi_{s-1}) \subseteq E(\pi_s)$ by Observation 2, we get $E(\pi_{s-1}) + e_s \subseteq E(\pi_s)$; which implies, by the adaptive monotonicity of $h$,

$$
\begin{aligned}
f_{avg}(\pi_s) - f(\pi_{s-1}) &= \mathbb{E}_{\Phi \sim \psi_{s-1}}[f_{avg}(\pi_s)] - h(\psi_{s-1}) \\
&\geq \mathbb{E}_{\Phi \sim \psi_{s-1}}[h(\psi_{s-1} + e_s)] - h(\psi_{s-1}) \\
&= \Delta(e_s \mid \psi_{s-1}). \qquad\qquad\square
\end{aligned}
$$

Note that we condition on the fact that the first $s-1$ steps of the policy $\pi$ are performed, therefore we have $f_{avg}(\pi_{s-1}) = h(\psi_{s-1})$. By taking expectation over all the possible realizations of the random variable $\Psi_{s-1}$ the lemma is proven.

**Lemma 4.** *Conditioned on any arbitrary partial realization $\psi$, we have $\mathbb{E}_{\Phi \sim \psi}[|E(\pi^*)|] \leq (k-1)d_{\text{in}}$.*

*Proof.* The optimal policy under each realization of the random variable $\Phi$ chooses at most $k$ items. Each one of these $k$ items (except the first one) will have at most $d_{\text{in}}$ incoming edges. Therefore, the expected number of edges is at most $(k-1)d_{\text{in}}$. $\qquad\square$

**Lemma 5.** *For every $1 \leq s \leq \ell$, we have*

$$\mathbb{E}_\Phi[h((E(\pi^*) \cap \mathcal{E}_{s-1}) \cup E(\pi_{s-1}))] \leq$$

$$\mathbb{E}_\Phi[h((E(\pi^*) \cap \mathcal{E}_s) \cup E(\pi_s))] + \frac{1}{\gamma} \cdot \mathbb{E}_\Phi[|E(\pi^*) \cap (\mathcal{E}_{s-1} \setminus \mathcal{E}_s)| \cdot \Delta(e_s \mid E(\pi_{s-1}))].$$

*Note that the expectation is taken over all the possible realizations of the random variable $\Phi$.*

*Proof.* The lemma follows by combining the two inequalities of Eq. (2) and Eq. (3).

$$\mathbb{E}_\Phi[\Delta(E(\pi^*)\cap\mathcal{E}_{s-1}\mid E(\pi_{s-1}))]-\mathbb{E}_\Phi[\Delta(E(\pi^*)\cap\mathcal{E}_s\mid E(\pi_{s-1}))] \tag{2}$$
$$=\sum\Pr[\Psi_{s-1}=\psi_{s-1}]\cdot\left[\mathbb{E}_{\Phi\sim\psi_{s-1}}[\Delta(E(\pi^*)\cap\mathcal{E}_{s-1}\mid\psi_{s-1})-\Delta(E(\pi^*)\cap\mathcal{E}_s\mid\psi_{s-1})]\right]$$
$$\overset{(a)}{\leq}\frac{1}{\gamma}\sum\Pr[\Psi_{s-1}=\psi_{s-1}]\cdot\mathbb{E}_{\Phi\sim\psi_{s-1}}[|E(\pi^*)\cap(\mathcal{E}_{s-1}\setminus\mathcal{E}_s)|\cdot\Delta(e_s\mid\psi_{s-1})]$$
$$=\frac{1}{\gamma}\cdot\mathbb{E}_\Phi[|E(\pi^*)\cap(\mathcal{E}_{s-1}\setminus\mathcal{E}_s)|\cdot\Delta(e_s\mid E(\pi_{s-1}))].$$

To see why inequality $(a)$ is true, note that for every given sub realization $\psi_{s-1}$ we have: (i) if $e_s$ is a dummy edge, then $(E(\pi^*)\cap\mathcal{E}_{s-1})\cup E(\pi_{s-1})=(E(\pi^*)\cap\mathcal{E}_s)\cup E(\pi_s)$, which makes $(a)$ trivial, or (ii) when $e_s$ is not dummy, $(a)$ results from Corollary 1.

$$\mathbb{E}_\Phi[h((E(\pi^*)\cap\mathcal{E}_{s-1})\cup E(\pi_{s-1}))]-\mathbb{E}_\Phi[h((E(\pi^*)\cap\mathcal{E}_s)\cup E(\pi_s))] \tag{3}$$
$$\leq\mathbb{E}_\Phi[h((E(\pi^*)\cap\mathcal{E}_{s-1})\cup E(\pi_{s-1}))]-\mathbb{E}_\Phi[h((E(\pi^*)\cap\mathcal{E}_s)\cup E(\pi_{s-1}))])$$
$$=\sum\Pr[\Psi_{s-1}=\psi_{s-1}]\cdot\mathbb{E}_{\Phi\sim\psi_{s-1}}[h((E(\pi^*)\cap\mathcal{E}_{s-1})\cup\psi_{s-1})-h(E(\pi^*)\cap\mathcal{E}_s)\cup\psi_{s-1})]$$
$$=\sum\Pr[\Psi_{s-1}=\psi_{s-1}]\cdot\mathbb{E}_{\Phi\sim\psi_{s-1}}[\Delta((E(\pi^*)\cap\mathcal{E}_{s-1})\mid\psi_{s-1})-\Delta(E(\pi^*)\cap\mathcal{E}_s)\mid\psi_{s-1})]$$
$$=\mathbb{E}_\Phi[\Delta(E(\pi^*)\cap\mathcal{E}_{s-1}\mid E(\pi_{s-1}))]-\mathbb{E}_\Phi[\Delta(E(\pi^*)\cap\mathcal{E}_s\mid E(\pi_{s-1}))]. \qquad\square$$

**Lemma 6.** $\mathbb{E}_\Phi[h((E(\pi^*)\cap\mathcal{E}_\ell)\cup E(\pi_\ell))]]\leq\dfrac{1}{\gamma}\cdot\mathbb{E}_\Phi[|E(\pi^*)\cap\mathcal{E}_\ell|\cdot\Delta(e_\ell\mid\Psi_{\ell-1})]+f_{avg}(\pi_\ell).$

*Proof.* We have

$$\mathbb{E}_\Phi[h((E(\pi^*)\cap\mathcal{E}_\ell)\cup E(\pi_\ell))-h(\pi_\ell)]$$
$$=\sum\Pr[\Psi_\ell=\psi_\ell]\cdot\mathbb{E}_{\Phi\sim\psi_\ell}[h(E(\pi^*)\cap\mathcal{E}_\ell)\cup\psi_\ell)-h(\psi_\ell)]$$
$$\overset{(a)}{\leq}\frac{1}{\gamma}\sum\Pr[\Psi_\ell=\psi_\ell]\cdot\mathbb{E}_{\Phi\sim\psi_\ell}[|E(\pi^*)\cap\mathcal{E}_\ell|\cdot\Delta(e_\ell\mid\psi_{\ell-1})]=\frac{1}{\gamma}\cdot\mathbb{E}_\Phi[|E(\pi^*)\cap\mathcal{E}_\ell|\cdot\Delta(e_\ell\mid\Psi_{\ell-1})].$$

To see why inequality $(a)$ is true, note that for every given sub realization $\psi_\ell$ we have: (i) if $e_\ell$ is a dummy edge, then $\mathcal{E}_\ell=\varnothing$, which makes inequality $(a)$ trivial, and (ii) if $e_\ell$ is not a dummy edge then we conclude inequality $(a)$ from the definition of weakly adaptive set submodular functions (see Definition 1).

The lemma follows by combining this inequality with the observation that $f_{avg}(\pi_\ell)=\mathbb{E}_\Phi[h(\pi_\ell)]$. $\quad\square$

To combine the last two lemmata, we need the following observation.

**Observation 3.** *For every* $2\leq s\leq\ell$, $\mathbb{E}_\Phi[\Delta(e_{s-1}\mid E(\pi_{s-2}))]\geq\gamma\cdot\mathbb{E}_\Phi[\Delta(e_s\mid E(\pi_{s-1}))].$

We are now ready to prove Theorem 1.

*Proof of Theorem 1.* Combining Lemmata 5 and 6, we get

$$f_{avg}(\pi^*)-f_{avg}(\pi_\ell)=\mathbb{E}_\Phi[h((E(\pi^*)\cap\mathcal{E}_1)\cup E(\pi_0))]-f_{avg}(\pi_\ell)$$
$$\leq\frac{1}{\gamma}\cdot\sum_{s=1}^\ell\mathbb{E}_\Phi[|E(\pi^*)\cap(\mathcal{E}_{s-1}\setminus\mathcal{E}_s)|\cdot\Delta(e_s\mid E(\pi_{s-1}))]$$
$$+\mathbb{E}_\Phi[\Delta((E(\pi^*)\cap\mathcal{E}_s)\cup E(\pi_s))]-f_{avg}(\pi_\ell)$$
$$\leq\frac{1}{\gamma}\sum_{s=1}^\ell\mathbb{E}_\Phi[|E(\pi^*)\cap(\mathcal{E}_{s-1}\setminus\mathcal{E}_s)|\cdot\Delta(e_s\mid E(\pi_{s-1}))]$$
$$+\frac{1}{\gamma}\cdot\mathbb{E}_\Phi[|E(\pi^*)\cap\mathcal{E}_\ell|\cdot\Delta(e_\ell\mid\Psi_{\ell-1})]$$

$$= \frac{1}{\gamma} \cdot \sum_{s=1}^{\ell-1} \mathbb{E}_\Phi[|E(\pi^*) \cap (\mathcal{E}_0 \setminus \mathcal{E}_s)| \cdot [\Delta(e_s \mid E(\pi_{s-1})) - \Delta(e_{s+1} \mid E(\pi_s))]]$$

$$+ \frac{1}{\gamma} \cdot \mathbb{E}_\Phi[|E(\pi^*) \cap \mathcal{E}_0| \cdot \Delta(e_\ell \mid E(\pi_{\ell-1}))], \quad (4)$$

where the first equality holds since the fact that $\sigma_0$ is an empty sequence implies $E(\sigma_0) = \varnothing$ and $\mathcal{E}_0 = E$, and the second equality holds since $\mathcal{E}_s \subseteq \mathcal{E}_{s-1}$ by Observation 2 for every $1 \leq s \leq \ell$. We now observe that for every $1 \leq s \leq \ell$, $\pi_s$ contains at most $2s$ vertices. Since each one of these vertices can be the end point of at most $d_{\text{in}}$ arcs, we get

$$|E(\sigma^*) \cap (\mathcal{E}_0 \setminus \mathcal{E}_s)| \leq |\mathcal{E}_0 \setminus \mathcal{E}_s| \leq 2s d_{\text{in}}$$

Additionally, by Lemma 4,

$$|E(\sigma^*) \cap \mathcal{E}_0| \leq |E(\sigma^*)| \leq (k-1)d_{\text{in}} \leq 2\ell d_{\text{in}}.$$

Plugging the last two inequalities into Inequality (4) yields

$$f_{avg}(\pi^*) - f_{avg}(\pi_\ell) \leq \sum_{s=1}^{\ell-1} \frac{2s d_{\text{in}}}{\gamma} \cdot \mathbb{E}_\Phi[\Delta(e_s \mid E(\pi_{s-1})) - \Delta(e_{s+1} \mid E(\pi_s))] +$$

$$\frac{2\ell d_{\text{in}}}{\gamma} \cdot \mathbb{E}_\Phi[\Delta(e_\ell \mid E(\sigma_{\ell-1}))]$$

$$= \sum_{s=1}^{\ell} \frac{2d_{\text{in}}}{\gamma} \cdot \mathbb{E}_\Phi[\Delta(e_s \mid E(\pi_{s-1}))] \leq \frac{2d_{\text{in}}}{\gamma} \cdot \sum_{s=1}^{\ell} [f_{avg}(\pi_s) - f_{avg}(\pi_{s-1})]$$

$$= \frac{2d_{\text{in}}}{\gamma} \cdot [f_{avg}(\pi_\ell) - f_{avg}(\pi_0)] \leq \frac{2d_{\text{in}}}{\gamma} \cdot f_{avg}(\pi_\ell),$$

where the second inequality holds due to Lemma 3 and the last inequality follows from the non-negativity of $f$. Rearranging the last inequality, we get

$$f_{avg}(\pi_\ell) \geq \frac{\gamma}{2d_{\text{in}} + \gamma} \cdot f_{avg}(\pi^*),$$

which implies the theorem since $f_{avg}(\pi_\ell)$ is a lower bound on the expected value of the output sequence of Algorithm 1 because $\sigma_\ell$ is always a prefix of this sequence. $\qquad\square$

## B.3 Proof of Theorem 2

In this section, we first restate and then prove Theorem 2 which guarantees the performance of our proposed policy applied to hypergraphs.

**Theorem 2.** *For adaptive monotone and weakly adaptive sequence submodular function $f$, the policy $\pi'$ represented by Algorithm 2 achieves*

$$f_{avg}(\pi') \geq \frac{\gamma}{r d_{\text{in}} + \gamma} \cdot f_{avg}(\pi^*),$$

*where $\gamma$ is the weakly adaptive submodularity parameter, $\pi^*$ is the policy with the highest expected value and $r$ is the size of the largest hyperedge in the input hypergraph.*

In the proof of this theorem we use the same notation that we used in Section B.2 for analyzing Algorithm 1, with the exception of $\mathcal{E}_s$, which is now defined as $\mathcal{E}_s = \{e \in E \mid \sigma_s \cap V(e) \text{ is a prefix of } e\}$, and $\ell$, which is now defined as $\lfloor k/r \rfloor$.

The following lemma is a counterpart of Lemma 4.

**Lemma 7.** $|E(\sigma^*)| \leq (k - r + 1)d_{\text{in}}$.

*Proof.* For a realization $\phi$, every arc of $\pi^*$ must end at a vertex of $\pi^*$ which is not one of the first $r - 1$ vertices. The observation follows since $\pi^*$ contains at most $k - r + 1$ vertices of this kind, and at most $d_{\text{in}}$ arcs can end at each one of them. $\qquad\square$

---

**Algorithm 2** Adaptive Hyper Sequence Greedy

---

1: **Require:** Directed hypergraph $H(V, E)$, $\gamma$-adaptive and adaptive-monotone function $h : 2^E \times O^E \to \mathbb{R}_{\geq 0}$ and cardinality parameter $k$
2: Let $\sigma \leftarrow ()$
3: **while** $|\sigma| \leq k - r$ **do**
4:      $\mathcal{E} = \{e \in E \mid \sigma \cap V(e) \text{ is a prefix of } e\}$
5:      **if** $\mathcal{E} \neq \varnothing$ **then**
6:          $e^* = \arg\max_{e \in \mathcal{E}} \Delta(e \mid \psi_\sigma)$
7:          **for** every v $\in e^*$ in order **do**
8:             **if** $v \notin \sigma$ **then**
9:                $\sigma = \sigma \oplus v$
10:            **end if**
11:          **end for**
12:          Identify the state of all edges in $\mathcal{E}' = \{e \in E \mid \text{all elements of } V(e) \text{ belong to } \sigma \text{ and appear in the same order}\}$
13:          $\psi_\sigma = \psi_{\mathcal{E}'}$
14:      **else**
15:          **break**
16:      **end if**
17: **end while**
18: **Return** $\sigma$

---

One can observe that the proofs of all the other observations and lemmata of Section B.2 are unaffected by the differences between Algorithm 1 and Algorithm 2, and thus, these observations and lemmata can be used towards the proof of Theorem 2.

*Proof of Theorem 2.* The proof of this theorem is identical to the proof of Theorem 1 up to two changes. First, instead of getting an upper bound of $2sd_{\text{in}}$ on $|\mathcal{E}_0 \setminus \mathcal{E}_s|$ for every $1 \leq s \leq \ell$, we now get an upper bound of $rsd_{\text{in}}$ on this expression because $\sigma_s$ might contain up to $rs$ vertices rather than only $2s$. Second, instead of getting an upper bound of $2\ell d_{\text{in}}$ on $|E(\sigma^*)|$, we now use Lemma 7 to get an upper bound of $(k - r + 1)d_{\text{in}} \leq r\ell d_{\text{in}}$ on this expression. $\qquad\square$

## C    Proof of Theorem 3

The approximability of the sequence submodular maximization, as a generalization of the densest $k$ subgraph problem (DkS) [32], is an open theoretical question with important implications. In this section, we prove Theorem 3.

In the DkS problem the goal is to find a subgraph on exactly $k$ vertices that contains the maximum number of edges. DkS as a generalization of the $k$-clique problem is NP-hard and the best polynomial algorithm for DkS achieves a $n^{1/4+\epsilon}$ approximation factor[3] for an arbitrary $\epsilon > 0$ [8]. Furthermore, there exists no polynomial time algorithm that approximates DkS within an $O(n^{1/(\log\log n)^c})$ factor unless 3-SAT has a subexponential time algorithm [40].

**Lemma 8.** *Any algorithm with an $\alpha$ approximation factor to the sequence submodular maximization problem solves the densest $k$ subgraph problem (DkS) with at most an $\alpha$ approximation factor.*

*Proof.* To prove this lemma, we show that for each instance of DkS over a directed graph $G(V, E)$ we can build an instance of the sequence submodular maximization problem over a directed graph $H(V, E')$ such that solving the latter problem also solves the former one. We assume all vertices and edges have a single state. Therefore, the problem translates to the non-adaptive sequence submodular scenario.

Graph $H$ is built from graph $G$ by replacing each edge $e = (u, v)$ in $E$ by two directed edges $(u, v)$ and $(v, u)$. We define $h(S) = |S|$, which is linear and therefore submodular. Finally, the sequence

610 submodular function $f$ is defined as $f(\sigma) = h(E(\sigma)) = |E(\sigma)|$. It remains to show that for every
611 subset of vertices $S$ the value of function $f$ for an arbitrary permutation $\sigma_S$ of $S$ is equivalent to
612 the size of subgraph $G_S$ induced by those vertices in graph $G$. This is true because for every edge
613 $(u, v) \in G_S$ we have two corresponding edges in the directed graph $H$ and based on the order of $u$
614 and $v$ exactly one of them is considered in $E(\sigma_S)$.

615 As a result, maximizing the function $f$ with a cardinality constraint $k$ is equivalent to solving the
616 DkS problem. Thus, any algorithm with an $\alpha$ approximation factor to the sequence submodular
617 maximization problem solves DkS with at least an $\alpha$ approximation factor. $\qquad\square$

618 Manurangsi [40] showed that any algorithm with an $O(n^{1/(\log \log n)^c})$ approximation factor to the
619 DkS problem (for a constant $c > 0$) would prove the exponential time hypothesis is false. Next, we
620 directly state the result of [40].

621 **Theorem 4** (Manurangsi [40], Theorem 1)**.** *There is a constant $c > 0$ such that, assuming the*
622 *exponential time hypothesis, no polynomial-time algorithm can, given a graph $G$ on $n$ vertices and a*
623 *positive integer $k \leq n$, distinguish between the following two cases:*

624 - *There exist $k$ vertices of $G$ that induce a $k$-clique.*

625 - *Every $k$-subgraph of $G$ has density at most $n^{-1/(\log \log n)^c}$.*

626 To sum-up, Theorem 3 is proved from the combination of the two following facts:

627 1. If there is an algorithm with an approximation within a $n^{1/(\log \log n)^c}$ factor to the sequence
628 submodular maximization problem, from the result of Lemma 7, we know that it would
629 solve the DkS problem with at most the same factor.

630 2. If there is an algorithm with a $n^{1/(\log \log n)^c}$ approximation factor to the DkS problem,
631 it could distinguish the two cases of Theorem 4 and would prove the exponential time
632 hypothesis to be false.

633 # D  Additional Experimental Details

634 ## D.1  Amazon Product Recommendation

635 In this application, we consider the task of recommending products to users. In particular, we use
636 the Amazon Video Games review dataset [41], which contains 10,672 products, 24,303 users, and
637 231,780 confirmed purchases. We furthered focused on the products that had been purchased at least
638 50 times each, leaving us with a total of 958 unique products.

639 Although we are using a different dataset, the experimental set-up closely follows that of the movie
640 recommendation task in Tschiatschek et al. [53] and Mitrovic et al. [44]. We first group and sort
641 all the data so that each user $u$ has an associated sequence $\sigma_u$ of products that they have purchased.
642 These user sequences are then randomly partitioned into a training set and a testing set using a 80/20
643 split. Note that we 5 trials to average our results.

644 Using the training set, we build a graph $G = (V, E)$, where $V$ is the set of all products and $E$ is the
645 set of edges between these products. Each product $i \in V$ has a self-loop $(i, i)$, where the weight
646 (denoted $w_{ii}$) is the fraction of users in the training set that purchased product $v_i$. Similarly, for each
647 edge $(i, j)$, the corresponding weight $w_{ij}$ is defined to be the conditional probability of purchasing
648 product $j$ given that the user has previously purchased product $i$.

649 For each sequence $\sigma_u$ in the test set, we are given the first $g$ products that user $u$ purchased, and then
650 we want to predict the next $k$ products that she will purchase. After each product is recommended
651 to the user, the state of the product is revealed to be 1 if the user has indeed purchased that product,
652 and 0 otherwise. At the start, the $g$ given products are known to be in state 1, while the states of the
653 remaining products are initially unknown.

654 As described in Section 2, the states of the edges are determined by the states of the nodes. In this
655 case, the state of each edge $(i, j)$ is equal to the state of product $i$. The intuitive idea is that edge $(i, j)$
656 encodes the value of purchasing product $j$ after already having purchased product $i$. Therefore, if

the user has definitely purchased product $i$ (i.e., product $i$ is in state 1), then they should receive the full value of $w_{ij}$. On the other hand, if she has definitely not purchased product $i$ (i.e., product $i$ is in state 0), then edge $(i, j)$ provides no value. Lastly, if the state of product $i$ is unknown, then the expected gain of edge $(i, j)$ is discounted by $w_{ii}$, the value of the self-loop on $i$, which can be viewed as a simple estimate for the probability of the user purchasing product $i$. See Figure 2a for a small example.

We use a probabilistic coverage utility function as our monotone adaptive submodular function $h$. Mathematically,

$$h(E_1) = \sum_{j \in V} \left[ 1 - \prod_{(i,j) \in E_1} (1 - w_{ij}) \right],$$

where $E_1 \subseteq E$ is the subset of edges that are in state 1.

## D.2 Wikipedia Link Prediction

We use the Wikispeedia dataset [56], which consists of 51,138 completed search paths on a condensed version of Wikipedia that contains 4,604 pages and 119,882 links between them. We further condense the dataset to include only articles that have been visited at least 100 times, leaving us with 619 unique pages and 7,399 completed search paths.

One natural idea for scoring each algorithm would be to look at the length of the shortest path between the predicted target and the true target. However, the problem with this metric is that all the popular pages have relatively short paths to most potential targets (primarily since they have so many available links to begin with). Hence, under this scoring, just choosing a popular page like "Earth" would be competitive with many more involved algorithms.

Instead, we define a measure we call the *Relevance Distance*. The relevance distance of a page $i$ to a target page $j$ is calculated by taking the average shortest path length to $j$ across all neighboring pages of $i$. A lower distance indicates a higher relevance. For example, if our target page is *Computer Science*, both *Earth → Earth Science → Computer Science* and *University → Education → Computer Science* have a shortest path of length 2. However, the relevance distance of *Earth* to *Computer Science* is 2.68, while the relevance distance of *University* to *Computer Science* is 2.41, which fits better with the intuition that *University* is logically closer to *Computer Science*.

## D.3 Deep Learning Baseline Details

### D.3.1 Feed Forward Neural Network

For both experiments, the input to the Feed Forward Neural Network is a size $|V|$ vector $X$. That is, there is one input for each item in the ground set. In the Amazon product recommendation task in Section 4.1, $X_i = 1$ if the user is known to have purchased product $i$ and 0 otherwise. Similarly, for the Wikipedia link prediction task in Section 4.2, $X_i = 1$ if the user is known to have visited page $i$ and 0 otherwise.

The output in both cases is a size $|V|$ soft-maxed vector $Y$. In Section 4.1, $Y_i$ can be viewed as the probability that product $i$ will be the user's next purchase. In Section 4.2, $Y_i$ can be viewed as the probability that user will visit page $i$ next.

For the Amazon product recommendation task in Section 4.1, each user $u$ in the training set has an associated sequence $\sigma_u$ of products she purchased. Each such sequence was split into $|\sigma_u| - 2$ training points by taking the first $g$ products as input and the $(g + 1)$-th product as the output for $g = 1, \ldots, |\sigma_u| - 1$. For each user $u$ in the testing set, we would take the first $g = 4$ products she purchased and encode them in the vector $X$ as described above. We would then input this vector into our trained network and output the vector $Y$. In the non-adaptive case we cannot get any feedback from the user, so we simply output the products corresponding to the $k$ highest values in $Y$.

In the adaptive case, we would look at the largest value $Y_j$ in our output vector and output this as our first recommendation. We then check if the corresponding product appeared somewhere later in the user's sequence $\sigma_u$. If yes, then we would update our input $X$ so that $X_j = 1$ and re-run the network to get our next recommendation. If not, we would simply use the next highest value in $Y_j$ as our next recommendation (since the input doesn't change). This was repeated for $k$ recommendations. This is supposed to mimic interaction with the user where we would recommend a product, and then see

whether or not the user actually purchases this product. Note that we only considered values $Y_j$ such that $X_j = 0$ because we did not want to recommend products that we knew the user had already purchased.

The main difference for the Wikipedia task in Section 4.2 is that, in the testing phase, we cannot simply output the top $k$ values in $Y$ as we did above because they likely will not constitute a valid path. Instead, we only have an adaptive version that is similar to what was described above. We find the highest value $Y_j$ such that $X_j = 0$ (i.e. the user had not already been to this page) and a link to page $j$ actually exists from our current page. We output this page $j$ as our recommendation for the user's next page. We then check if the user actually visited our predicted page $j$ at some point in their sequence of pages. If yes, we would update $X$ so that $X_j = 1$ and re-run the network. If not we would look to the next highest value in the output $Y$. This was repeated for $k$ guesses. Note that if we reached the true target page, we would stop making guesses.

In terms of architecture, we used a single hidden layer of 256 nodes with ReLU activations. We use a batch size of 1024 at first and then go down to a batch size of 32 when we are in the low data regime (i.e. only using 1% of the available training data). We used an 80/20 training/validation split to guide our early stopping criterion during training (with minimum improvement of 0.01 and patience of 1). We used categorical cross-entropy as our loss function.

### D.3.2 LSTM

The main difference between the LSTM and the feed forward network is in the input. The input to the LSTM is a sequence of one-hot encoded vectors instead of just a single vector. That is, for the LSTM, each vector in the sequence had exactly one index with value 1.

We experimented with using a long sequence of input vectors and padding with all-zero vectors, but we found better results using a fixed small sequence length $g$ and then "pushing" the sequence back when updating. For example, if our current input was a sequence of vectors $[v_1, v_2, v_3]$ and we wanted to update it with a new vector $v_4$, the updated input would be $[v_2, v_3, v_4]$.

The adaptive LSTM followed the same set-up as the non-adaptive LSTM, but with the same adaptive update rules described above for the feed-forward neural network.

For all experiments, we used a single hidden layer of 8 LSTM nodes. The other hyperparameters are all the same as described for the Feed Forward network above, except we start at a batch size of 256 instead of 1024 (before also going down to a batch size of 32 in the low data regime).