[Reviews · NeurIPS 2019]

Reviewer 1



I think that their approximation guarantees are a solid theoretical contribution in the submodular optimization space and I think that this is an interesting theoretical problem to consider. I also like that they have lower bounds showing that a dependence on the degree is necessary. Further the authors were able to improve results from (Mitrovic et al., 2018) (the authors should feature these results for prominently). One thing the authors should point out is that their greedy algorithm requires calculating an expectation over all realizations. In practice, this would be difficult to do. To improve clarity, the authors should formally write out the optimization problem that they want to solve. The fact that there is a cardinality constraint is just quickly mentioned in the text and the pseudocode. It should be written more prominently (preferably the problem statement should have its own definition block). I think the hardness of approximation results should be n^(-1/(log log n)^C), so that it is a number less than 1. The authors also write "we develop a novel analysis technique to guarantee the performance of our algorithms." The authors should discuss more about this novel analysis technique. From the proof it is not completely clear the interesting points. The experiments in the paper are sufficient. However, I don't believe that this approach would be significant in a practical setting where there are user / item features and the distribution of realizations is unknown. I am not surprised that LSTMs do not perform well in the experiments the authors performed, as the experiments were almost purely combinatorial optimization problems. The authors mention the probabilistic coverage utility function as being weakly submodular. What is the weak submodularity constant? *** Post Rebuttal *** Thank you for your rebuttal. It would be good to include the (maybe heuristic) derivation on bounding the weak submodularity constant in the paper.

Reviewer 2



The paper introduces a new concept of submodularity, which is called the adaptive sequence submodularity defined on directed graphs and hypergraphs. The sequence submodularity on directed graphs and hypergrahs has been already proposed for non-adaptive maximization by Tschiatschek et al. (AAAI17) and Mitrovic et al. (AISTATS18). Moreover, extensions of submodularity to adaptive optimization has been studied well for various concepts of submodularity. However, such an extension is not known for sequence submodularity. The paper is the first study to do it. The adaptive sequence submodularity has many applications obviously, and thus I think the problem studied in this paper is very important. The paper presents an adaptive greedy algorithm for maximizing the adaptive sequence submodular monotone functions. This guarantee is better by a factor 1-1/e even compared with the existing guarantees given by Tschiatschek et al. (AAAI17) and Mitrovic et al. (AISTATS18) for the non-adaptive sequence submodular functions. Because of this, the significance of this theoretical guarantee is also obvious. The authors also verified the performance of the algorithms through experiments whose settings are realistic. Summing up, the paper introduces a new useful concept of submodularity, which has obviously many applications, and gives a new theoretical guarantee of a greedy adaptive algorithm improving upon the existing guarantee for special cases. These contributions are valuable both from the modeling and the theoretical aspects. Update: After reading the author's rebuttal, I have chosen to maintain my score.

Reviewer 3



Though the proposed framework is novel and general, I have some doubt about its formulation: the user's feedback/realization is revealed after each element is selected seems to make the sequence selection problem easier and less useful in practice. Since we get immediate feedback, the problem becomes that given the history of a user's reaction to a sequence of elements, select the next best element. Or in other words, under the current setting, the algorithm cannot generate a sequence of recommendations on its own, while in practice, for example, we often need to recommend a sequence of movies/ads for users to select/click rather than one at a time. Moreover, it is not intuitively clear to me why the function is modeled as weekly submodular except that a bound can be obtained based on the gamma parameter. In addition, even though there is a theoretical guarantee, the gamma value can be infeasible to compute depending on the function and thus we may not be able to know the guarantee in practice. For the experiment part, it is quite reasonable that the proposed method can outperform the baselines that don't utilize the immediate user feedback. Because of the immediate feedback, the feedforward network seems to suit the task very well, which can give good performance if there is enough data. The current method only outperforms DNN based methods when there are few training data, which again limits the usefulness of the approach. Additionally, I also question if the current method can scale to large dataset computationally. The paper is well organized and the writing is clear. After rebuttal: Thanks for the detailed feedback from the authors. I have read all other reviewers' comments. My concern about the batch setting and why the function is modeled weakly-submodular remains. Particularly, for the weakly-submodular part, I was expecting some explanations or examples of why the problem is close to submodular. Otherwise, the weakly-submodular setting is merely a tool for getting a bound for optimizing an arbitrary function. I keep my evaluations unchanged.

[Author Response · NeurIPS 2019]

We thank the reviewers for their constructive feedback. We first address the general comments and then answer specific questions.

**GENERAL COMMENTS**

Reviewers 1 and 2 both enquired about the novelty of the proof techniques in this paper. In a nutshell, the original proof from
Mitrovic et al. (2018) could not be extended to (weak) adaptivity because it was not linear enough to allow for the necessary and
extensive use of the linearity of expectation. Our proof replaces several of their lemmas with tighter, more linear analyses (e.g.,
Lemma 5 in our paper replaces Lemma 3.3[1]). We agree that a high-level summary would be useful for clarity, so we will add a
paragraph in the main paper summarizing the techniques we used to achieve the necessary linearity, and the resulting benefits.

Reviewers 1 and 3 brought up valid questions about the computational feasibility of this approach in practice. It is true that if the
distribution of realizations is difficult to calculate, then our approach may struggle. Theoretically speaking, an approximation of
the expected values which is in a $1 \pm \varepsilon$ range of the true value introduces only a $(1 - \varepsilon)$ factor in our guarantees. Furthermore,
many real-world applications can utilize efficient and accurate approximations. For example, in our first experiment, the "true"
distribution over a user's product preferences is unknown. Instead, we approximate the conditional probabilities simply by
counting the number of times a shopper has purchased product $i$ after product $j$. In fact, as the attached code shows, the runtime
of our approach is superior to the deep learning methods. In the revised version, we can add graphs comparing the runtimes. We
also note that there are several works on speeding-up greedy algorithms (e.g., stochastic greedy), which can help us scale our
approach to much larger datasets.

**REVIEWER 1**

**"I think the hardness of approximation results should be $n^{\frac{-1}{(\log\log n)^c}}$, so that it is a number less than 1."**
Thank you for noting this, we will fix it. We wanted to be consistent with the paper we were citing, but you are correct that this is
inconsistent with the rest of our paper.

**"What is the weak submodularity constant [of the probabilistic coverage utility function]?"**
It depends on the data, but roughly speaking, it is inversely proportional to the smallest edge weight in the underlying graph.

**". . . this type of problem (no user features or item features) is not where LSTMs are known to shine."**
This is a fair point and we will add it as a remark in the paper. We would like to note that our approach can also take advantage
of user/product features (as a part of the distribution over realizations), but the data did not include any additional features.

**"To improve clarity, the authors should formally write out the optimization problem that they want to solve."**
**". . . authors were able to improve results from (Mitrovic et al., 2018) . . . feature these results more prominently."**
We appreciate the reviewer's suggestions for improving the clarity of the paper, and we will incorporate these changes.

**"Improvements: If the authors could tighten the approximation guarantees..."**
We have hope that it is possible to remove the 2 from the current approximation factor of $\frac{\gamma}{2d_{\text{in}}+\gamma}$, but we believe that achieving a
sub-linear dependence on the degree would require a totally different proof approach.

**REVIEWER 2**
**"In line 110, the authors explain how the state of an edge is decided, but this part is not clear to me."**
The basic idea of our framework is that the state of each edge is determined entirely by the state of its endpoints. It may be easier
to think about it as a function that takes in the states of the two endpoint vertices and outputs the state of the edge. One key
advantage of our framework is that it works with *any* such function. For example, in our experiments, the state of each edge is
assigned to be the same as the state of its start vertex. We will clarify this point further in the revised version.

**REVIEWER 3**
**"I think the framework needs to be extended so that a set/sequence of elements is selected at each step . . . "**
Generally speaking, the point of adaptive submodularity is to select items one at a time and receive feedback, so our theoretical
analysis followed this standard. That being said, we agree that in practice it is common to have to select multiple items before
receiving feedback. Our approach can be easily adjusted to do so by simply delaying the observation of the states of the selected
vertices (i.e., waiting to update the conditional distribution). This idea can also be linked to the recently studied problem of
submodular maximization via "adaptive rounds", where in each round one picks multiple items in parallel [5, 6, 18]. It must be
noted that new theoretical analysis would be required for this change, but we agree that batch selection for weakly adaptive
sequence submodularity would be an interesting direction for future work.

**". . . it is not intuitively clear to me why the function is modeled as weakly submodular . . . "**
There are two main reasons for using weakly submodular functions. First, weakly submodular functions can model a much
greater variety of problems (Khanna et al., 2017; Elenberg et al.,2017). Second, weak submodularity is strictly more general.
Intuitively, $\gamma$ tells us how close the problem is to being submodular and when $\gamma = 1$ we recover regular submodularity. We
want to emphasize that we do not need to know $\gamma$ to run our algorithm, but we do agree that $\gamma$ can sometimes be infeasible to
calculate, and thus, we may not know the guarantee we are getting in practice.

**"The current method only outperforms DNN based methods when there are few training data . . . "**
Given enough data, neural network-based approaches show state-of-the-art performance for most tasks, so we did not expect
our method to achieve superior accuracy in such scenarios. Instead, our approach provides numerous other advantages such as
theoretical guarantees, interpretability, ease of implementation, and robustness (while still maintaining comparable accuracy).
In the second experiment, we saw that even when we use all available data, it is still not enough for the deep learning-based
approaches, which shows that the robustness against data scarcity is a practically significant aspect of our method.

## Footnotes

[1]As numbered in the version of their paper appearing in `https://arxiv.org/pdf/1802.09110.pdf`.


[Meta-Review · NeurIPS 2019]

The paper introduces a new concept of submodularity, which is called the adaptive sequence submodularity defined on directed graphs and hypergraphs. The model is interesting and has a wealth of important applications. The propose an algorithm with strong theoretical guarantees. The analysis in the adaptive setting is not trivial and there are new and interesting ideas introduced. Furthermore, the authors show that this algorithm has strong guarantees for special cases of the problem studied in the past. In addition, the authors perform a series of experiments on interesting applications, showing their effectiveness in practice. This is a well written paper, which introduces a new model that captures important applications, new algorithmic techniques, improves on special cases from previous work, and performs rigorous experiments. We therefore strongly recommend on accepting this paper.